Bacterial community profiles within the water samples of leptospirosis outbreak areas

Md Lasim Asmalia 1 2 asmaliaccb@gmail.com
Mohd Ngesom Ahmad Mohiddin 3
http://orcid.org/0000-0002-2132-2346 Nathan Sheila 2
http://orcid.org/0000-0002-3045-7963 Abdul Razak Fatimah 2
http://orcid.org/0000-0002-1453-1238 Abdul Halim Mardani 4
Mohd-Saleh Wardah 1
Zainul Abidin Kamaruddin 5
Mohd-Taib Farah Shafawati 2 farah_sh@ukm.edu.my
1 Department of Herbal Medicine Research Centre, Insitute for Medical Research , Setia Alam , Malaysia
2 Department of Biological Science and Biotechnology, Universiti Kebangsaan Malaysia , Bangi , Malaysia
3 Centre for Communicable Research, Insititute for Public Health , Setia Alam , Malaysia
4 Biotechnology Research Institute, Universiti Malaysia Sabah , Kota Kinabalu , Malaysia
5 Faculty of Applied Science, Universiti Teknologi MARA (UiTM) Cawangan Pahang , Bandar Tun Razak, Pahang , Malaysia
Grohmann Elisabeth
Electronic publication date: 2024 Apr 29
Publication date: 2024
Volume: 12
Electronic Location ID: e17096
Received 2023 Oct 29; Accepted 2024 Feb 21
Copyright: © 2024 Md Lasim et al.
Copyright year: 2024
Copyright holder: Md Lasim et al.
License: This is an open access article distributed under the terms of the Creative Commons Attribution License, which permits unrestricted use, distribution, reproduction and adaptation in any medium and for any purpose provided that it is properly attributed. For attribution, the original author(s), title, publication source (PeerJ) and either DOI or URL of the article must be cited.
License URL: https://creativecommons.org/licenses/by/4.0/

Keywords: Bacterial community, Leptospirosis, Pathogenic, Next generation sequencing, Public health, Water recreational, Leptospira

Funding: Fundamental Research Grant Scheme (FRGS) Ministry of Higher Education FRGS/1/2018/STG03/UKM/02/1 National Institute of Health, Ministry of Health Malaysia This research was funded by the Fundamental Research Grant Scheme (FRGS), Ministry of Higher Education FRGS/1/2018/STG03/UKM/02/1 and National Institute of Health, Ministry of Health Malaysia. The funders had no role in study design, data collection and analysis, decision to publish, or preparation of the manuscript.

==============================
Background

Leptospirosis is a water-related zoonotic disease. The disease is primarily transmitted from animals to humans through pathogenic Leptospira bacteria in contaminated water and soil. Rivers have a critical role in Leptospira transmissions, while co-infection potentials with other waterborne bacteria might increase the severity and death risk of the disease.

Methods

The water samples evaluated in this study were collected from four recreational forest rivers, Sungai Congkak, Sungai Lopo, Hulu Perdik, and Gunung Nuang. The samples were subjected to next-generation sequencing (NGS) for the 16S rRNA and in-depth metagenomic analysis of the bacterial communities.

Results

The water samples recorded various bacterial diversity. The samples from the Hulu Perdik and Sungai Lopo downstream sampling sites had a more significant diversity, followed by Sungai Congkak. Conversely, the upstream samples from Gunung Nuang exhibited the lowest bacterial diversity. Proteobacteria, Firmicutes, and Acidobacteria were the dominant phyla detected in downstream areas. Potential pathogenic bacteria belonging to the genera Burkholderiales and Serratia were also identified, raising concerns about co-infection possibilities. Nevertheless, Leptospira pathogenic bacteria were absent from all sites, which is attributable to its limited persistence. The bacteria might also be washed to other locations, contributing to the reduced environmental bacterial load.

Conclusion

The present study established the presence of pathogenic bacteria in the river ecosystems assessed. The findings offer valuable insights for designing strategies for preventing pathogenic bacteria environmental contamination and managing leptospirosis co-infections with other human diseases. Furthermore, closely monitoring water sample compositions with diverse approaches, including sentinel programs, wastewater-based epidemiology, and clinical surveillance, enables disease transmission and outbreak early detections. The data also provides valuable information for suitable treatments and long-term strategies for combating infectious diseases.

Introduction

Leptospirosis is the most prevalent zoonotic disease belonging to various species in the spirochete genus Leptospira interrogans (Pan American Health Organization (PAHO), 2015). Currently, 64 Leptospira species, including pathogenic, intermediate, and saprophytes variants, have been recorded (Vincent et al., 2019). Diagnosing leptospirosis is complex, as the disease is commonly asymptomatic and mimics other diseases. Leptospirosis patients typically experience sudden fever and headaches, while some display non-specific symptoms that could overlap with several pulmonary hemorrhagic and Weil’s diseases.

Leptospirosis has become a significant public health concern in urban and suburban populations, particularly in tropical and subtropical regions. Approximately over a million people are affected, and 60,000 fatalities are attributable to the infectious disease yearly (Costa et al., 2015). The prevalence of Leptospirosis also varies by region, ranging from 0.5/100,000 to 95/100,000 human population in Europe and Africa, respectively (Levett & Haake, 2015).

Although Leptospira could remain viable between several weeks and months in flowing water, the organism could only survive for several weeks in stagnant water (Bierque et al., 2020; Yanagihara et al., 2022). Environmental factors, including heavy rains, flooding, hurricanes, and global changes, have been associated with significant Leptospirosis outbreaks (Hacker et al., 2020; Naing et al., 2019). Heavy rainfalls and floods are also considered the primary risk for leptospirosis infections (Chadsuthi et al., 2021; Hacker et al., 2020).

Pathogenic and saprophytic Leptospira strains could be washed into water bodies from soil (Evangelista & Coburn, 2010). Consequently, humans residing in flood-prone and contaminated areas have increased exposure risk to infections (Radi et al., 2018). Leptospira could also infect humans via abrasion, cuts, or any liquids directly exposed to the contaminated sources, primarily during physical or occupation-related activities, such as swimming, recreation, agriculture, mining, forestry, fishing, and aquaculture.

Leptospirosis infections post-exposure to the contaminated sources might depend on the bacterial ability to survive and establish infection in new hosts. The compositions of the bacterial community influence the survival and persistence of Leptospira in the environment (Barragan et al., 2017). Consequently, the situation could lead to co-infections with other pathogenic bacteria, worsening the disease (Mohd Ali et al., 2017; Sapian et al., 2012).

Co-infections could complicate diagnosis and prolong examination and mistreatment, potentially leading to mortality (Lasim et al., 2021). Furthermore, the sensitivity and specificity of Leptospira and other environmental bacterial communities might be affected by chemical, physical, or biological contaminations. Low environmental leptospires could also produce false-negative results. Consequently, microbial diversity studies could enable new species identification that might impact co-infection occurrences.

The next-generation sequencing (NGS) technique has revolutionised bacterial community structural analyses, establishing bacterial population variations in specific samples (Rawat & Joshi, 2019). The method facilitates biodiversity, metabolomics, and metabolic pathway assessments by utilising 16S rDNA, including partial 16S rDNA amplification, eDNA methods, protein, whole genome sequencing, shotgun, Methyl-Seq, and RNA-Seq (Satam et al., 2023). The NGS approach has also streamlined microbial evaluations, making them more efficient, highly accurate, and cost-effective (Kanzi et al., 2020). The technique also possesses the ability to generate millions of short-read sequences more rapidly than Sanger sequencing (Kanzi et al., 2020). The present study aimed to determine the bacterial community profiles in selected leptospirosis-endemic areas. The results could contribute to a better comprehension of the microbiome ecosystem.

Materials and Methods

Ethical note

The present study obtained a wildlife handling permit from Jabatan Perlindungan Hidupan Liar dan Taman Negara (PERHILITAN). The Universiti Kebangsaan Malaysia Animal Ethical Committee (UKMAEC) has also approved this study, FST 2016 SHUKOR/18-MAY/750-MAY 2016-SEPT-2018-AR-CAT2.

Research area and sampling

The current study conducted water sampling at three waterfall areas in Hulu Langat, Selangor; Hulu Perdik (HP) (03°12′18.1″N and 101°50′10.0″E)), Sungai Congkak (SC) (03°12′35.7″ N and 101°50′10.0″ E), Sungai Lopo (SL) (03°13′08.6″N and 101°51′52.5″ E), and Gunung Nuang (GN) (03°12′54.1″ N and 101°53′02.1″ E) (Fig. 1). The HP sampling site was chosen due to the leptospirosis outbreak 2016, where approximately 12 cases were reported (Neela et al., 2019). On the other hand, the SC area was identified by the Health Department as a potential leptospirosis area (Azhari et al., 2018).

Figure 1 The water sampling locations in four recreational areas, Hulu Langat, Selangor; (A) HP, (B) SC, (C), SL, and (D) GN.

Map were generated using ArcGIS 10.5 (Esri, Redlands, CA, USA) with the base maps was digitized from OpenStreetMap (http://www.openstreetmap.org) under CC BY-SA 2.0

Most sampling sites record an average of approximately 1,800 mm yearly rainfall and 25 °C annual temperature. The selected areas are categorised as recreational forests and located near forest reserves. The sampling locations are commonly packed during weekends and public holidays, with visitors performing outdoor activities, such as bathing, camping, fishing, and swimming.

Approximately 5 L of water samples from each of the three independent upstream, midstream, and downstream points were collected in triplicates and poured into sterile bottles. The upstream sites were selected for their natural cleanliness, while the downstream and midstream points were approximately 200 to 500 m from areas with frequent recreational activities. The water samples were collected in January and December 2018. The samples were immediately stored in a cooler box at 4 to 6 °C and transported to the laboratory within 1 to 3 h before further analysis.

DNA extraction

The 5 L water samples obtained from each sampling area were filtered with a Milipore Glass base vacuum filtration system to concentrate the water. Subsequently, DNA extractions were performed with a Machenery Nagel Nucleospin® Soil Genomic DNA extraction kit (Macherey-Nagel GmbH & Co. KG, Germany) according to the manufacturer’s instructions. The water samples were then transferred into NucleoSpin® Bead tubes containing ceramic beads before being homogenised by vortexing for 5 min.

The water samples were lysed with SL 1 and SX buffers. The proteins in the samples were denatured and removed with Buffer SB before being washed with a NucleSpin® Soil column. The purity and concentration of the extracted DNA were assessed with a spectrophotometer (Implen NanoPhotometer® N60/N50). Subsequently, fluorometric quantification was performed with iQuant™ Broad Range dsDNA Quantification Kit. Further quality control was conducted via gel electrophoresis to visualise the genomic DNA bands.

16S rDNA sequencing

In this study, the purified genomic DNA obtained was employed to generate a 16S rRNA library. The primers utilised were 5′TCGTCGGCAGCGTCAGATGTGTATAAGAGACAGCCT ACGGGNGGCWGCAG-3′ and 5′GTCTCGTGGGCTCGGAGATGTGTATAAGAGACAGGA CTACHVGGGTATCTAATCC-3′. The underlined nucleotides were the prokaryotic conserved 16S rRNA loci-specific V3 and V4 domains, while the rest were the Illumina overhang adapter sequences. The present study performed paired-end sequencing with Illumina MiSeq (2 × 300) sequencing platform, and all sequencing task were outsourced to 1st BASE (Apical Scientific Sdn Bhd, Seri Kembangan, Malaysia). The raw reads and sequences obtained were then submitted to the National Centre for Biotechnology Information (NCBI) Sequences Read Archive (SRA) database with the BioProject accession number PRJNA987398.

Data analysis

A bioinformatic software was employed to process and analyse the data obtained in the current study. Adaptor sequences and low-quality reads were filtered with BBDUK of the BBTools package. Subsequently, the sequences were merged with USEARCH before being matched with the 16S rRNA. Any chimeric errors were removed with VSEARCH. This study compared the operational taxonomic units (OTUs) and alpha and beta diversity assignments by comparing the QIIME against the silver database. Statistical analyses were performed with Statistical Package for the Social Sciences (SPSS), where p < 0.05 indicated significant differences with R studio along the package ranges, including vegan, ggplot, metacoder, Phyloseq, and Krona.

Results

Sequencing

Low-quality and short-sequence reads obtained in the present study were trimmed. Consequently, The 12 water samples generated 1,194,806 raw sequences. The GN samples recorded the highest abundance with 109,121 sequences, followed by HP with 98,097, SC at 96,059, and SL with 94,989.

Bacterial community richness and diversity indices

Alpha diversity

The alpha diversities of the samples in this study were calculated according to Chao1, Shannon, and Simpson indices. The results revealed that the richness and diversity of the species were varied among the samples (Fig. 2). The SC samples documented the highest Chao1 and Simpson levels, suggesting that the site possessed richer bacterial species than the other locations. On the other hand, the SL water sampless exhibited the most remarkable diversity based on Simpson indices, which were significantly different to GN. Nonetheless, no considerable difference was recorded between the HP and SC samples. The observation indicated that the dominant bacterial species produced similar effects on the overall population within the areas. Conversely, significant differences were documented by the samples collected from HP, SC, SL, and GN according to Shannon and Simpson indices (p < 0.05).

Figure 2 The alpha diversity (Chao1, Shannon, and Simpson diversity indices) boxplots based on OTUs abundance in the water samples calculated with analysis of variance (ANOVA) (pairwise analysis).

The vertical axis denotes richness estimations in the number of OTUs, separate boxes are overlaid on study site combinations, and asterisks indicate significant differences: *p < 0.05, **p < 0.01, and ***p < 0.001.

The rarefaction curve generated in the current study was employed to assess the sequencing depth and bacterial community richness of the samples. The samples from SL and SC demonstrated higher bacterial communities than the HP and GN samples. The results indicated that the data collected had sufficient to describe the bacterial community diversity in the sampling areas (Fig. 3A).

Figure 3 The alpha diversity of the bacterial communities in the water samples; the (A) rarefaction curves illustrate OTUs species richness in each area and (B) Venn diagram demonstrating the numbers of unique and shared OTUs between sampling areas based on 16S rR.

The OTUs were obtained with QIIME V1.9.1. with a 97% sequence similarity, overlapping colours indicate shared OTUs, non-overlapping colours represent unique OTUs, and all OTUs are defined at a 97% level of sequence similarity.

The Venn diagrams illustrated in Fig. 3B were utilised to analyse bacterial communities in the water sampless, focusing on unique and shared operational taxonomic units (OTUs). The results revealed 14 shared OTUs between all samples. On the other hand, the HP sample featured 692 unique OTUs, SC recorded 648, SL had 522, and GN documented 52. The SC and HP samples shared 750 OTUs, while 250 OTUs were shared by SL and HP water samples. SL water samples documented the highest uniqueness, while GN exhibited the lowest OTUs.

The water sampless evaluated in the present study recorded significant species richness, albeit with considerable variations. The comprehensive microbial community species in each sampling site was visualised with a Krona graph plotted utilising the Krona Tool. A different colour in the figure indicates different bacteria and their ratio in the samples.

In the samples collected from HP, the class identified were Gammaproteobacteria (48%), Alphaproteobacteria (27%), Acidobacteria (6%), Vicinamibacteria (1%), Verrucomicrobiae (5%), Actinobacteria (2%), Acidimicrobiia (1%), Bacteroidia (2%), Planctomycetes (2%), Campylobacteria (1%), Nitrospiria (0.4%), Elusimicrobia (0.3%), Methylomirabiia (0.2%), Fusobacteriia (0.09%) Gemmatimonadetes (0.09%), WPS-2 (0.03%), NB1-j (0.03%), Latescibacterota (0.01%) and FCPU426 (0.2%) (Fig. 4A). The bacteria found in the samples collected from SL belonged to the class Gammaproteobacteria (52%), Alphaproteobacteria (13%), Acidobacteriae (5%), Vicinamibacteria (4%), Verrucomicrobiia (5%), Actinobacteria (2%), Acidimicrobiia (1%), Thermoleophilia (3%), Bacteroidia (3%), Planctomycetes (3%), Nitrospiria (1%), Camphylobacteria (0.3%), Elusimicrobia (0.2%), FCPU426 (0.4), Methyylomirabilia (0.4%), NB1-j (a.2%), Lastescibacterota (0.01%) and Spirochaetia (0.03%) (Fig. 4B). The water samples procured from SC documented Gammaproteobacteria (38%), Alphaproteobacteria (18%), Acidobacteriae (7%), Vicinamibacteria (2%), Holophagae (2%), Verrucomicrobiae (4%), Actinobacteria (2%), Acidimicrobiia (1%), Thermoleophilia (1%), Bacteroidia (9%), Planctomycetes (2%), Campylobacteria (2%), Nitrospiria (1%), Myxococcia (2%), Spirochaetia (1%), Elusimicrobia (0.5%), FCPU426 (0.03%), Methylomirabilia (1%), Gemmatimonadetes (0.3%), and Latescibacteria (0.02%) (Fig. 5A). Nonetheless, in the samples collected from GN, Bacilli (89%), Gammaproteobacteria (11%), Campylobacteria (0.03%), Nitrospiria (0.007%) and Gemmatimonadetes (0.03%) were dominant (Fig. 5B). An interactive version of the chart is available on the Krona tools.

Figure 4 The Krona graph demonstrating the relative abundance of taxonomic hierarchy and percentage composition in phyla, class, family, order and species levels based on OTUs abundances in the (A) HP, (B) SL, water samples.

The samples from HP and SL had a significant relative abundance of Gammaproteobacteria, Alphaproteobacteria, Acidobacteria and Verrucomicrobiae.

Figure 5 The Krona graph demonstrating the relative abundance of taxonomic hierarchy and percentage composition in phyla, class, family, order and species levels based on OTUs abundances in the (A) SC, and (B) GN water samples.

The samples from SC and GN had a significant relative abundance of Gammaproteobacteria.

Beta diversity

The present study employed beta diversity to assess dissimilarities between each sample. The principle coordinate analysis (PCoA) based on Bray-Curtis distances was employed during the assessment. Visualisations of the bacterial community variations provided an understanding of bacterial structural patterns in the water samples. The principle components that reflected the different bacterial community compositions in the samples were expressed through the horizontal (PC1), vertical (PC2), and horizontal (PC3) axes (Anderson et al., 2011).

Based on the OTUs levels, PCoA recorded 96.26% bacterial community composition variation (PC1-77.28%, PC2-11.32%, and PC3-7.66%), indicating a clear separation among each sampling area (Fig. 6A). The GN samples documented a distinct cluster from the other clusters in HP, SL, and SC. The data revealed that intra-sample similarity was higher among the samples from SC, SL, and HP than GN, which might be influenced by environmental factors.

Figure 6 The beta diversity results of the water samples; the (A) PCoA between the HP, SL, SC, and GN sampling sites based on OTUs abundance (weighted UniFrac distance) and (B) heatmap based on the distance matrix analysis with weighted UniFrac.

The variance explained by each principle coordinate axis is denoted by PC1 vs. PC2 and PC1 vs. PC3. Colour hues represent the distance matrix; a darker hue indicates significant abundance, while a lighter colour represents low abundance. A high abundance reflected a significant gap relationship between samples.

Unweighted UniFrac (UUF) offers valuable information on taxa status, particularly rare taxa, where a small value indicates less variation in species diversity between two samples. The unweighted pair group method with arithmetic mean (UPGMA) tree heatmap obtained in this study revealed that the UUF-based sample distances ranged between 0.0373 and 0.0931. Generally, the UniFrac distance data illustrated relatively minor differences in the bacterial communities among the species in the water samples. Furthermore, the bacterial communities exhibited considerable likeness, probably due to the proximity of the sample collection sites (Fig. 6B).

Bacterial community compositions

The bacterial species distribution in each water sample was identified at the phylum, genus, and species levels (Fig. 7). Proteobacteria, Firmicutes, Actinobacteria, Acidobacteria, and Bacteroidota were the dominant phyla in all samples. Specifically, Proteobacteria was the most abundant bacterial phylum in the samples.

Figure 7 The relative abundance of the major 10 bacteria at the (A) phyla, (B) genera, and (C) species levels based on OTUs abundance obtained from sites 1, 2, and 3 in HP, SL, SC, and GN.

Water samples from HP comprised approximately 67.87%, 9.26%, 8.43%, and 3.98% Proteobacteria, Acidobacteriota, Cyanobacteria, and Actinobacteria, respectively. The samples collected from SL documented 73.51%, 8.09%, 6.38% and 3.95% Proteobacteria, Acidobacteriota, Actinobacteriota, and Verruomicrobiota, respectively. Proteobacteria (72.33%), Acidobacteriota (6.57%), Actinobacteriota (6.36%) Bacteroidota (5.25%) were found in SC samples, whereas GN water samples contained Firmicutes (90.97%), Proteobacteria (8.89%), Bacteroidota (0.06%), and Actinobacteriota (0.03%) (Fig. 7A).

At the genus level, HP water samples recorded a remarkable abundance of Chloroplast (24.61%), Curvibacter (24.60%), Pseudomonas (21.74%), uncultured (17.37%), and Novosphingobium (10.54%). Varying bacterial genera were also documented by SL and SC water samples despite similar abundant genera, Pseudomonas and uncultured bacteria, which contributed over 25%. On the other hand, the relative abundance of Curvibacter and Cadecea in the samples was 7.67% and 18.11%, respectively. The GN water samples were dominated by Bacillus (90.31%) and Herbaspirillum (9.29%) (Fig. 7B). The sample also recorded Curvibacter lanceolatus (25.36%) as the most abundant bacterial species, followed by uncultured organism (15.32%), Planoglabratella opercularis (7.86%), Serratia marcescens (13.22%), and Chromobacterium rhizoryzae (11.25%) (Fig. 7C).

In this study, the species most recorded were C. lanceolatus, uncultured organism, Planoglabratella opercularis, S. marcescens, and Chromobacterium rhizoryzae. Curvibacter lanceolatus was the most abundant species in SL, followed by SC, GN, and HP. Remarkably, S. marcescens was also documented in all sampling sites, which is an opportunistic pathogenic to human.

The present study applied heatmap analysis to determine the relative abundance differences between the top 40 genera contained in all water samples (Fig. 8). The result could be divided into three primary clusters, and for every nine samples collected from HP (HP1, HP2, HP3), SC (SC1, SC2, SC3), and SL (SL1, SL2, SL3) were laid in the two subclusters with slight variations. Conversely, the water samples from GN (GN1, GN2, and GN3) were classified as a single cluster. Unweight trees also revealed that only the bacterial communities from GN were in a similar group. The bacterial communities in SP, SL, and SC exhibited more significant similarity to each other and were placed within a broader cluster range compared to GN.

Figure 8 The heatmap and hierarchical cluster results based on the relative abundance of bacterial communities in the water samples.

The numbers indicate the sample number of the corresponding sampling site, the x-axis represents similarities or dissimilarities between the sampling areas, the y-axis denotes similarities or dissimilarities between the genera, dark blue represents the more abundant genus, and light blue corresponds to the genus with less abundance in each sample.

Discussion

This study could contribute valuable insights into the bacterial community diversity of leptospirosis endemic areas in Hulu Langat, Malaysia. Furthermore, the coexistence of Leptospires with other bacteria could lead to co-infection in humans. Consequently, the present study aimed to provide an overview of bacterial community profiles in water samples from four recreational areas.

The Ministry of Health (MOH) Malaysia defines a leptospirosis outbreak as more than one probable or confirmed case with an epidemiological link within one incubation period (Ministry of Health Malaysia, 2011). In 2016, a leptospirosis outbreak was reported in HP. Neela et al. (2019) suggested that the outbreak might be due to the site being employed for military training. Furthermore, tourists visit the area for its scenic beauty (Neela et al., 2019). Although no cases have been recorded in SL and SC, the sites were considered hotspots due to frequent suspected cases reported in the areas.

Proteobacteria, Firmicutes, and Acidobacteria were the predominant phyla in all water samples evaluated in this study which was also reported by Bai et al. (2014) and Wei & Pan (2015). Similar findings were reported by El-Chaktoura et al. (2015) and Atnafu, Desta & Assefa (2022), where the water samples were collected from the Meuse River, the Netherlands, and water supplies in Ethiopia, respectively. The results of the current study highlighted the crucial requirement for enhanced diagnostic tools to effectively identify newly discovered pathogenic bacteria and obtain a deeper understanding of their role in disease transmission.

Ahmad et al. (2021) discovered that Proteobacteria tends to be the predominant phylum affected by human activities in urban freshwater lakes. Human activities alter the composition, elevate the anaerobic decomposers, and increase the organic contents of the water. Proteobacteria is a gram-negative phylum bacteria consisting of various classes and families. The microorganism is the most abundant and commonly found in ecosystems, including water, soil, animal niches, and plants (Rizzatti et al., 2017). Krishna et al. (2020) noted the essential role of the bacteria in complex biogeochemical processes.

Firmicutes was the second most abundant phylum in the water samples assessed in the present study. The bacteria is commonly encountered in wastewater with elevated antibiotic levels and extreme environmental conditions (Fang et al., 2017). The phylum is the only one known for being endospore-forming (Fang et al., 2017). Firmicutes are also one of the most widespread groups of bacteria due to their remarkable resilience and dispersal capabilities. Takarina, Sukma & Adiwibowo (2022) documented a significantly higher abundance of Firmicutes in contaminated areas, such as mining sites, showcasing its impressive tolerance to heavy metals. Lin et al. (2019) also reported the predominance of Firmicutes in urban and suburban sediments in the Yangtze River Delta of China. The study observed a significant negative correlation.

The current study is among the first attempts to explore bacterial communities in recreational areas by applying NGS techniques. This study focused on four recreational park areas (HP, SL, SC, and GN). Nevertheless, no pathogenic Leptospira group were detected in the samples. The absence of pathogenic Leptospira in the samples indicated its inability to survive in flowing water, particularly rivers or streams. The results also indicated viable Leptospira scarcity, suggesting that rivers are not playing an important role in the leptospirosis transmissions.

The ability of pathogenic Leptospira to survive in water varies (Saito et al., 2013). For instance, some findings demonstrated that viable pathogenic Leptospira strains were still detectable nine weeks after the initial contamination or index case (Thibeaux et al., 2017). The results in the present study aligned with the limited detection of Leptospira in aquatic ecosystems. Most of the positive samples in this study were obtained from stagnant water or contained deficient concentrations (Schreiber et al., 2020; Saito et al., 2013).

Elevated levels of potentially pathogenic bacteria from the Burkholderiales order were detected in the water samples assessed in this study, where the Curvibacter genus was the most predominant. The microorganism could be found in various environments, such as soil and water (Vuronina et al., 2015). Moreover, certain species have resisted common antibiotics (Vuronina et al., 2015). Currently, only a few recognised species within the Curvibacter, including C. gracilis, C. delicatus, C. lanceolatus, and C. fontanus, are the mesophilic bacteria typically found in aquatic surroundings (Ding & Yokota, 2010; Patel et al., 2022; Ziganshina et al., 2016). Ziganshina et al. (2016) also reported the prevalence of Curvibacter bacteria in patients with atherosclerotic plaque. The microorganisms are also considered pathogens in lungs and cystic fibrosis patients (Card et al., 2010).

S. marcescens was identified in the samples evaluated in this study. The presence of the bacteria from the Serratia genus highlighted microbial co-infection possibilities. S. marcescens is an aerobic, gram-negative bacillus. The microorganism is an opportunistic human pathogen with a broad host range. The bacteria have been associated with various infections, including urinary and respiratory tracts, wounds, septicemia, eyes, and bloodstream infections (Khanna, Khanna & Aggarwai, 2012). Co-infections involving S. marcescens with other microbes, such as Escherichia coli, Pseudomonas aeruginosa, and Candida glabrata, have also been reported in diabetic patients (Sharifipour et al., 2020).

Nori et al. (2020) noted bacterial co-infections in under 5% of coronavirus 2019 (COVID-19) patients from March to April 2020, raising concerns for vulnerable patients. In another report, documented that 20 of 43 bacterial cultures obtained from 60 patients had S. marcescens and P. aeruginosa. Skedros et al. (2014) recorded a unique case of a diabetic individual with multiple comorbidities with a C. glabrata and S. marcescens co-infection. In another study, Bazaid et al. (2022) emphasised the strong correlations between considerably antibiotic-resistant S. marcescens and COVID-19.

Antimicrobial therapy-based S. marcescens treatments exhibited variable outcomes due to the resistance among the bacterial species. Increasing mortalities and resistance incidences in patients resulted in the recommendation of empirical treatments with drugs employment, including aminoglycosides, tazobactam, carbapenems, and cephalosporins, where the final therapy would be guided by culture sensitivity (Prakash et al., 2021).

The aquatic bacterial community compositions between downstream HP, SC, SL, and upstream GN sampling sites documented significant differences. The variations were primarily linked to the activities occurring in each area. The HP, SL, and SC are popular water recreation sites, thus human interactions and domestic activities are higher than at the GN sampling areas (Winde, 2010).

Significant levels of human interactions and activities lead to enhanced microbial diversity due to slower water flow and easy access to water collection (Traore et al., 2016). Howard et al. (2003) found higher pathogenic bacteria contamination levels in downstream rivers than those with minimal human activity. Additional factors, such as wastewater treatment plants, stormwater runoff, and non-disinfected septic tank systems, also contribute to the increased presence of bacteria in the environment (Hong, Qiu & Liang, 2010). Furthermore, substantial organic matter in sediments could create new environmental conditions supporting the proliferation of non-native species, influencing specific phyla dominance at different sampling locations (Al et al., 2022; Luo et al., 2020).

A primary limitation of the present study is the sampling period. Most water samples were collected in 2018, 2 years after the reported leptospirosis outbreak. The temporal gap might have affected the detection of pathogenic Leptospira in the water samples, as the presence of the pathogens could have fluctuated over the period. Another limitation pertains to potential sample alterations during storage and transportation. The changes could introduce biases in DNA and microbiological samples, impacting the findings accuracy (Takahara, Minamoto & Doi, 2015).

A third limitation is to identify the presence of potential non-viable microorganisms in the samples. The cultivation-independent methods employed in this study frequently lack details of microbial cell physiological conditions, including whether they are dormant, stressed, or in an active growth phase. Assessing microorganism viability commonly requires insights into their physiological status (Acharya et al., 2020). Nevertheless, obtaining the data solely through genetic analysis is challenging (Acharya et al., 2020).

Chimeric sequence risks during assembly and potential incomplete extension in one cycle of mispriming necessitate further validation of the results (Gaspar & Thomas, 2013). Consequently, longer reads might be required. Nevertheless, the approach might increase cost and reduce read quality under some conditions (Ho et al., 2021). Future research should address the limitations to enhance the robustness and accuracy of the findings.

Conclusions

The upstream and downstream water samples evaluated in this study significantly differed in bacterial community diversity and richness. The findings highlighted the impact of geographic distance. The downstream areas exhibited a substantially increased diversity, richness, and abundance of potentially pathogenic bacteria. The variations were probably influenced by human and animal activities, underscoring the pivotal role of the activities in regulating waterborne environmental health.

The current study identified potential pathogenic bacterial species, that could infect the visitor and subsequently causing co-infection in these particular sites. The findings also critically implicate developing strategies to prevent environmental contaminations by pathogenic bacteria and reduce the risk of co-infection with other diseases. Nonetheless, acknowledging the limitations of sequencing-based metagenomic studies is crucial. Further studies should aim to unravel the relationships between bacterial communities, environmental factors, and human health.

The authors would like to thank the Director General of Health Malaysia for his permission to publish this article. The gratitude is also extended to the Wildlife Ecology and Disease team (Rosha, Zahin, and Farisha).

Additional Information and Declarations

Competing Interests

Author Contributions

Field Study Permissions

Data Availability

The authors declare that they have no competing interests.

Asmalia Md Lasim conceived and designed the experiments, performed the experiments, analyzed the data, prepared figures and/or tables, authored or reviewed drafts of the article, and approved the final draft.

Ahmad Mohiddin Mohd Ngesom performed the experiments, analyzed the data, prepared figures and/or tables, authored or reviewed drafts of the article, and approved the final draft.

Sheila Nathan conceived and designed the experiments, authored or reviewed drafts of the article, and approved the final draft.

Fatimah Abdul Razak conceived and designed the experiments, analyzed the data, authored or reviewed drafts of the article, and approved the final draft.

Mardani Abdul Halim analyzed the data, authored or reviewed drafts of the article, and approved the final draft.

Wardah Mohd-Saleh performed the experiments, authored or reviewed drafts of the article, and approved the final draft.

Kamaruddin Zainul Abidin analyzed the data, prepared figures and/or tables, and approved the final draft.

Farah Shafawati Mohd-Taib conceived and designed the experiments, analyzed the data, authored or reviewed drafts of the article, and approved the final draft.

The following information was supplied relating to field study approvals (i.e., approving body and any reference numbers):

This project has obtained permit from PERHILITAN on wildlife handling as well as approved by the UKM Animal Ethical Committee (UKMAEC) with approval no. (FST 2016 SHUKOR/18-MAY/750-MAY 2016-SEPT-2018-AR-CAT2).

The following information was supplied regarding data availability:

The sequences are available at NCBI GenBank: PRJNA987398.

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
