# Peer review of "Bacterial community profiles within the water samples of leptospirosis outbreak areas"

_PeerJ, doi:10.7717/peerj.17096_

## Round 0.1 · original submission · Major Revisions

I agree with the opinion of the reviewers. The topic of your work is interesting. However, several modifications/clarifications as well as improving the quality of your figures and careful language checks throughout the manuscript are mandatory to make your work even more understandable and readable. Thus, please carefully address the comments made by both reviewers.

**Language Note:** The review process has identified that the English language must be improved. PeerJ can provide language editing services - please contact us at [email protected] for pricing (be sure to provide your manuscript number and title). Alternatively, you should make your own arrangements to improve the language quality and provide details in your response letter. – PeerJ Staff

·

Basic reporting

Dear authors,

thank you for your article “Bacterial communities profiles within the water samples of Leptospirosis outbreak areas” and your work in this field. The background of your study is clearly stated both in abstract and introduction. Moreover, the topic is set into the context of leptospirosis and the relevance of this disease. The article is well structured and raw data is deposited at NCBI.
However, I have a couple of comments on your manuscript that you may find below:

1a. There are many -yet mostly minor- grammar and spelling mistakes. English correction should be conducted to improve readability and clarity of the text. Here are some corrections/suggestions to start with:
Line 39: were instead of was
Line 41: please rephrase the last part of the sentence “Thus reduced…”
Line 53: the most prevalent/ one of the most prevalent/ a prevalent
Line 53: variety of species
Line 59: a major public health concern
Lines 50-51: It was estimated that each year more than…
Line 63: what do you mean by population? People?
Line 65: humans
Line 66: What do you mean by “occupation-related water”?
Line 67: factors, hurricanes
Line 74: may depend/depends
Line 81: prolong, lead to mistreatment
Line 84: specificity
Line 86: low level
Line 90: Please rephrase or complete the sentence “the ability to generate millions of short-read sequences in short time”
Line 91: Gel Electrophoresis
Line 127: concentrate (instead of concrete)?
Line 130: nuclease-free water
Line 151: quality and quantity
Line 172: trimmed
Line 182-183: …indices were used to estimate the richness, which exhibited variations across the samples?
Line 199: Additional? Meaning 750+250?
Lines 255-256: doubling of “reported”
Line 263: a gram-negative phylum?
Line 292: transmission

1b. The resolution of all figures is too low. Please provide figures in a resolution that allows to read the data provided. Besides, it is important to increase the font size of numbers and graph labels/descriptions so that it is possible to read it without enlarging the document.

1c. Line 210: Please describe the main findings depicted in Fig. 5a. In my view Sungai Lopo is distinct from the other locations, which is an interesting finding.

1d. Line 221: What do you mean by abundance in this context?

1e. Line 245: What do you mean by clade in this case?

Experimental design

The research question is clear and the article aims to contribute to research within a relevant field. The major limitation in my view is the long time between leptospirosis outbreaks and sampling. Therefore, it is not possible to trace back the previous outbreaks and identify the respective source of infection. However, limitations of the study are addressed in the discussion and the community analyses might be interesting as a reference in further studies.

However, I have some comments on the MM chapter that you may find below:

2a. The MM section on 16S rDNA Gene Sequencing is hard to understand and some information are missing. Please improve the readability and provide further information to allow others groups to repeat your experiments: composition of each mastermix, template DNA in each PCR, PCR programs.

2b. Please provide information on the buffer (SL1 or SL2 +/- enhancer) and the mode of mechanical disruption during DNA extraction (line 125ff).

2c. Line 144: first PCR with primer overhang? In my understanding the first PCR must have been conducted with the ordinary primers, while the overhang was "added" using a second PCR.

2d. Thank you for submitting your sequencing results to NCBI. Please also provide information on where to find your raw data in your MM section.

2e. Was the sequencing conducted in your lab? Please clarify.

2f. Are coordinates for the sampling points available? If so, please include them in the chapter on “Research area and sampling”.

Validity of the findings

3a. The conclusion 372 seems to be an overreach. Please either exclude this statement or further explain, how your findings contribute to early disease transmission and outbreak detection.

Additional comments

4a. Line 35: Down-/upstream of what? Please explain briefly.

4b. Line 91f.: In my understanding DGGE, Sanger, RFLP do not belong to NGS. Please revise this paragraph.

4c. Lines 100-102: Probably belongs to the Ethical Note?

4d. Line 121: “-“ or “to”

4e. Lines 229ff.: As a suggestion: maybe use brackets after the phyla for the percentages, as you did in the next paragraph (instead of using equal signs in the text). This would increase the readability and the style would be consistent throughout your manuscript.

4f. I do not understand how the attractiveness of the area to tourists affects the lack of reported leptospirosis. Please clarify.

4g. Line 310: As far as I am aware, S. marcescens was not mentioned in the results section. Please include the discussed finding in the results.

4h. Line 351: In this study only cultivation-independent methods were applied. Please explain why the detection of non-viable microbes would be more challenging.

4i. Fig. 2: Doubling of “were”

Reviewer 2 ·

Basic reporting

The article needs to be checked thoroughly for language and grammar. There are several spelling mistakes and typing errors throughout the manuscript.

Some figures are not readable because of low resolution. Please upload high-resolution images and make the axes clear.

Experimental design

No comment

Validity of the findings

No comment

Additional comments

1. Figures 2 and 6 need a figure caption.
2. Figure 4 is very difficult to read and the names on the figure are not clear. Please include the interpretation of results in the caption, briefly.
3. For the figures with alpha- and beta-diversity, please mention the following figure captions, on which region (16S V4) were the analyses performed, with how many reads, what the figures depict, and what each element in the figures indicates.
4. Line 185 mentions that as per the results using Chao1 and Shannon indices, Sungai Lopo showed the highest diversity, however, figure 2 shows Sungai Congkak to have the highest diversity with Chao1 and Shannon. As per Simpson, Sungai Lopo showed the highest diversity.
5. Please include an explanation for PC1, 2, and 3 in Figure 5a and explain the difference between the left and the right panels.

---

## Round 0.2 · accepted · Accept

Dear authors, both reviewers are satisfied with your careful revision of the manuscript. I am delighted to tell you that your manuscript has been accepted for publication.

Kind regards
Elisabeth Grohmann

·

Basic reporting

The manuscript was improved and my comments were adressed. The major critisism of the language and the resolution of the images was implemented. The manuscript in the current version is clear and well understandable and the figure resolution is sufficient.

Experimental design

All my comments were adressed. In my view, the experimental design is now clearly described.

Validity of the findings

My comment was adressed. I do not have any additional comments.

Additional comments

My comments were adressed. I do not have any additional comments.

Reviewer 2 ·

Basic reporting

The authors studied the bacterial communities in the water samples in Malaysia. The study focuses on Leptospirosis which is a relevant disease in tropical and subtropical areas. The figures and text have been revised as per the comments in the first revision.

Experimental design

The research question is well-defined and the study aims to determine the bacterial community profiles in selected leptospirosis-endemic areas. The authors also discuss the limitations of the study.

Validity of the findings

No comment

Additional comments

Were positive and negative controls used for Leptospira? If not, it would have been a good addition to the other samples. The sequencing technology used may struggle with accurately identifying low-abundance species.